# Association of Pulmonary Function Decline over Time with Longitudinal Change of Glycated Hemoglobin in Participants without Diabetes Mellitus

**DOI:** 10.3390/jpm11100994

**Published:** 2021-09-30

**Authors:** Wen-Hsien Lee, Da-Wei Wu, Ying-Chih Chen, Yi-Hsueh Liu, Wei-Sheng Liao, Szu-Chia Chen, Chih-Hsing Hung, Chao-Hung Kuo, Ho-Ming Su

**Affiliations:** 1Department of Internal Medicine, Kaohsiung Municipal Siaogang Hospital, Kaohsiung Medical University, 482 Shan-Ming Rd., Hsiao-Kang Dist., Kaohsiung 812, Taiwan; cooky-kmu@yahoo.com.tw (W.-H.L.); u8900030@yahoo.com.tw (D.-W.W.); 990329kmuh@gmail.com (Y.-C.C.); Liuboy17@gmail.com (Y.-H.L.); casablansa198711@gmail.com (W.-S.L.); scarchenone@yahoo.com.tw (S.-C.C.); pedhung@gmail.com (C.-H.H.); kjh88kmu@gmail.com (C.-H.K.); 2Division of Cardiology, Department of Internal Medicine, Kaohsiung Medical University Hospital, 482 Shan-Ming Rd., Hsiao-Kang Dist., Kaohsiung 812, Taiwan; 3Faculty of Medicine, College of Medicine, Kaohsiung Medical University, 482 Shan-Ming Rd., Hsiao-Kang Dist., Kaohsiung 812, Taiwan; 4Research Center for Environmental Medicine, Kaohsiung Medical University, 482 Shan-Ming Rd., Hsiao-Kang Dist., Kaohsiung 812, Taiwan; 5Division of Pulmonary and Critical Care Medicine, Department of Internal Medicine, Kaohsiung Medical University Hospital, 482 Shan-Ming Rd., Hsiao-Kang Dist., Kaohsiung 812, Taiwan; 6Division of Nephrology, Department of Internal Medicine, Kaohsiung Medical University Hospital, 482 Shan-Ming Rd., Hsiao-Kang Dist., Kaohsiung 812, Taiwan

**Keywords:** non-diabetes, glycated hemoglobin, pulmonary function, forced vital capacity, forced expiratory volume in 1 s

## Abstract

Pulmonary damage and function impairment were frequently noted in patients with diabetes mellitus (DM). However, the relationship between lung function and glycemic status in non-DM subjects was not well-known. Here, we evaluated the association of longitudinal changes of lung function parameters with longitudinal changes of glycated hemoglobin (HbA1c) in non-DM participants. The study enrolled participants without prior type 2 DM, hypertension, and chronic obstructive pulmonary disease (COPD) from the Taiwan Biobank database. Laboratory profiles and pulmonary function parameters, including forced vital capacity (FVC) and forced expiratory volume in 1 s (FEV1), were examined at baseline and follow-up. Finally, 7055 participants were selected in this study. During a mean 3.9-year follow-up, FVC and FEV1 were significantly decreased over time (both *p* < 0.001). In the multivariable analysis, the baseline (unstandardized coefficient β = −0.032, *p* < 0.001) and longitudinal change (unstandardized coefficient β = −0.025, *p =* 0.026) of FVC were negatively associated with the baseline and longitudinal change of HbA1c, respectively. Additionally, the longitudinal change of FVC was negatively associated with the risk of newly diagnosed type 2 DM (*p =* 0.018). During a mean 3.9-year follow-up, our present study, including participants without type 2 DM, hypertension, and COPD, demonstrated that the baseline and longitudinal change of FVC were negatively and respectively correlated with the baseline and longitudinal change of HbA1c. Furthermore, compared to those without new-onset DM, participants with new-onset DM had a more pronounced decline of FVC over time.

## 1. Introduction

Type 2 Diabetes mellitus (DM) is a globally common disease and a major risk factor of cardiovascular mortality and morbidity [1]. Micro-vascular and macro-vascular complications in particular organs and tissues, such as heart, kidney, and retina, are well established in patients with type 2 DM [2]. The lung is another target of organ damage of DM, and pulmonary function decline is also frequently noted in patients with type 2 DM [3]. According to recent studies, hyperglycemia is associated with pulmonary functional damage in type 2 diabetic patients with chronic obstructive pulmonary disease (COPD) and asthma [4,5,6]. Pulmonary function deterioration is associated with elevated glycated hemoglobin (HbA1c), glucose variability, and abnormal glucose level in type 2 diabetic patients with hospitalization for COPD or asthma [5,6,7,8]. In addition, anti-glycemic control in patients with type 2 DM can improve their pulmonary function [9].

Standard pulmonary function parameters, such as forced vital capacity (FVC) and forced expiratory volume in 1 s (FEV1), are generally used to assess the pulmonary function in clinical practices [10]. Compared with type 2 diabetic patients with well glycemic control, type 2 diabetic patients with poor glycemic control had a lower predicted FVC, FVC, predicted FEV1, and FEV1 [7,11]. Similarly, compared with non-diabetic subjects, patients with type 2 DM also had a lower residual FVC and residual FEV1 [12]. Although pulmonary function decline and status of glycemic control were studied in type 2 diabetic participants, there were few studies investigating the relationship between pulmonary function decline over time and longitudinal change of glucose status in non-DM participates. Here, we studied the relationship between changes of pulmonary function over time and longitudinal change of HbA1c in subjects without type 2 DM.

## 2. Methods

### 2.1. Data Source

This longitudinal study data was analyzed from the Taiwan Biobank, a national database with adults aged from 30 to 70 years old without cancer in Taiwan since 2008 [13,14]. The demographic information, including age, gender, and history of hypertension and diabetes, was obtained from a face-to-face interview. Systolic blood pressures, body mass index, and overnight fasting blood chemistry parameters, including fasting blood glucose, total cholesterol, triglycerides, serum creatinine, glutamic-pyruvic transaminase (GPT), and HbA1C, were collected. Initially, we included 27,210 participants that had completed all above data at baseline and at around a 3.9-year follow-up. Then, we excluded participants with type 2 DM (n = 1449), hypertension (n = 3562), COPD (n = 5952), and restrictive lung disease (n = 1). We also excluded participants who could not complete standard pulmonary function tests at baseline and longitudinal follow-up (n = 9191). Finally, 7055 participants were included in the present study.

### 2.2. Ethics Statement

The study protocol was approved by the Institutional Review Board of Kaohsiung Medical University Hospital (KMUHIRB-E(I)-20180242), which was approved on 8 March 2018.

### 2.3. Spirometry Examinations

Well-trained technicians performed the standard spirometry measurements by using a MicroLab spirometer and Spida 5 software (Micro Medical Ltd., Rochester, Kent, UK) [15]. All participants received a pulmonary function test three times, including FEV1 and FVC. All tests meet standard quality criteria with differences within 5% or 100 mL, and the best result among the 3 pulmonary function tests was used for further analysis. FVC-predicted and FEV1-predicted were calculated by dividing the FVC and FEV1 by the counterpart values measured from the general population by spirometry software, respectively. All participants did not receive a bronchodilator test.

### 2.4. Definition of Type 2 DM

Type 2 DM is defined according to the guidelines of the American Diabetes Association [16]. Participants were considered to have newly-diagnosed type 2 DM if type 2 DM was recorded by face-to-face interview or their HbA1C was ≥6.5% at follow-up.

### 2.5. Statistical Analysis

All data were presented as mean ± standard deviation or number (percentage). The longitudinal change of laboratory data and pulmonary function tests between baseline and follow-up were analyzed by paired *t*-test. The difference of continuous variables between groups was analyzed by an independent sample *t*-test. Correlations of baseline HbA1c and longitudinal change of HbA1C were determined by linear regression analysis. Binary logistic regression analysis was used to determine the significant correlations of newly diagnosed type 2 DM. Significant variables in the univariable linear and logistic analyses were selected into multivariable linear regression analysis with stepwise selection and logistic regression analysis with forward selection, respectively. All statistical analyses were performed by SPSS 22.0. A *p*-value of less than 0.05 was considered to be statistically significant.

## 3. Results

The present study has 7055 participants enrolled (mean age 49.5 ± 10.1, male gender 35%, smoking 25.6%) without type 2 DM, hypertension, or chronic pulmonary diseases during a 3.9-year follow-up. Table 1 shows clinical, laboratory, and pulmonary function data for all 7055 study participants. Compared to the baseline variables, age, systolic blood pressure, body mass index, fasting blood glucose, HbA1c, total cholesterol, triglyceride, FVC-predicted, FVE1-predicted, and FVE1/FVC were significantly increased and FVC and FEV1 were significantly decreased at follow-up. In addition, baseline FEV1 and FVC were 3.05 ± 0.59 and 3.64 ± 0.67 L for men and 2.09 ± 0.43 and 2.49 ± 0.48 for women, respectively. The annular decline of FEV1 and FVC were 19.5 ± 97.0 and 56.7 ± 93.2 mL for all participants, respectively.

The univariable and multivariable linear regression analyses of baseline HbA1c are shown in Table 2. In univariable analysis, baseline HbA1c was significantly correlated with male gender, increased age, systolic blood pressure, body mass index, fasting blood glucose, GPT, total cholesterol, and triglyceride and decreased FVC and FEV1. In multivariable analysis, baseline HbA1c was significantly correlated with increased age, body mass index, fasting blood glucose, GPT, total cholesterol, and triglyceride and decreased of FVC (unstandardized coefficient β = −0.033, 95% confidence interval [CI], −0.043 to −0.024; *p* < 0.001).

Univariable and multivariable linear regression analyses of the longitudinal change of HbA1c are shown in Table 3. In univariable analysis, the longitudinal change of HbA1c was positively correlated with the longitudinal changes of age, body mass index, fasting blood glucose, GPT, total cholesterol, and triglyceride and negatively correlated with longitudinal changes of creatinine, FVC, and FEV1. In multivariable analysis, the longitudinal change of HbA1c was positively correlated with the longitudinal changes of body mass index, fasting blood glucose, total cholesterol, and triglyceride and negatively correlated with longitudinal changes of creatinine and FVC (unstandardized coefficient β = −0.025, 95% CI, −0.048 to −0.003, *p =* 0.026).

Univariable and multivariable linear regression analyses of longitudinal change of FVC are shown in Table 4. In univariable and multivariable analysis, the longitudinal change of FVC was negatively correlated with smoking, and the longitudinal changes were positively correlated with age, GPT, and HbA1c (in multivariable analysis, unstandardized coefficient β = −0.022, 95% CI, −0.044 to −0.001, *p =* 0.041).

Table 5 shows the comparison of longitudinal changes of clinical, laboratory, and pulmonary function data between participants with and without newly diagnosed type 2 DM. Compared to participants without newly diagnosed type 2 DM, participants with newly diagnosed type 2 DM had increased longitudinal changes of fasting blood glucose and HbA1c and decreased longitudinal changes of total cholesterol and FVC during the 3.9-year follow-up.

Table 6 shows the odds ratio of the longitudinal changes of parameters in association with newly diagnosed type 2 DM in univariable and multivariable logistic regression analyses. In the multivariable analysis, newly diagnosed type 2 DM was positively correlated with the longitudinal change of fasting blood glucose and negatively correlated with the longitudinal changes of total cholesterol and FVC (odds ratio = 0.625, 95% CI, 0.424 to 0.922, *p =* 0.018).

## 4. Discussion

Our present study was the first one to demonstrate the relationship between the longitudinal changes of HbA1c and pulmonary function parameters in participants without type 2 DM, hypertension, and COPD. We showed that baseline HbA1c was associated with baseline FVC and the longitudinal change of HbA1c was also associated with the longitudinal change of FVC. Furthermore, compared to those without new-onset DM, participants with new-onset DM had a more pronounced decline of FVC over time.

The annular decline of pulmonary function was correlated with an ethnic difference, age, gender, smoke habit, diabetes, hypertension, and COPD [17,18,19,20,21], so we only enrolled participants without prior diabetes, hypertension, and COPD. There were some population-based studies to evaluate the annular decline in FEV1 and FVC. A 15-year Doetinchem population-based cohort study showed the annular decline in FEV1 and FVC were 36.1 and 59.9 mL for men and 30.9 and 29.0 mL for women, respectively [18]. In Korea, a 3.8-year cohort study enrolled 8842 health subjects without prior pulmonary diseases showed the annular decline in FEV1 was 31.3 mL for men and 27.0 mL for women [22]. In Korea, another 8.95-year longitudinal study, which consisted of 35,129 health subjects without a history of pulmonary diseases, demonstrated the annular decline in FEV1 was 37.32 mL [23]. In our 3.9-year follow-up study, the annular decline in FEV1 and FVC were 19.5 and 56.7 mL for all participants, 30.0 and 70.7 mL for men, and 13.8 and 49.2 mL for women, respectively. Different from our study, the above previous studies did not exclude participants with a history of hypertension and diabetes [18,22,23], which might partially explain why the annular decline in FEV1 was relatively low in our present study.

Diabetic lung disease was an emerged issue for pulmonary organ damage and function impairment in the DM population. Compared to subjects with non-diabetes, significant lung function impairments were frequently found in patients with type 2 DM. In the Atherosclerosis Risk in Communities Study, compared with non-diabetic subjects, patients with type 2 DM had a significantly lower FEV1 (3.2 vs. 3.4 L for men; 2.3 vs. 2.5 L for women) and FVC (4.2 vs. 4.6 L for men; 3.0 vs. 3.3 L for women) and faster annular decline in FEV1 (49 vs. 47 mL) and FVC (64 vs. 58 mL) [24]. From the Korea National Health and Nutrition Survey database, compared with non-DM participants, type 2 DM had a significantly lower FEV1 (2.7 vs. 2.8 L) and FEV1/FVC (75.9 vs. 77.9%) [25]. The difference between DM and non-DM patients for the FEV1/FVC ratio was insignificant in a pooled meta-analysis [26]. The baseline FEV1 (3.05 L for men; 2.09 L for women) and FVC (3.64 L for men; 2.49 L for women) in our participants were similar to those in non-diabetic and healthy Asian adults [25,27]. Compared to those with non-newly diagnosed type 2 DM, our subjects with newly diagnosed type 2 DM similarly had a reduced FVC but comparable longitudinal changes of FEV1 and FEV1/FVC.

High glycemic status, in terms of high HbA1c, was associated with pulmonary dysfunction and increased risks of hospitalization for asthma and COPD in patients with type 2 DM [5,28,29]. In a cross-sectional study in Japan, a 1% increase in HbA1c was associated with a significant decrease in FEV1 and FVC (73 and 128 mL for men and 25 and 52 mL for women, respectively) [28]. Although HbA1c was negatively correlated with FEV1 and FVC in DM patients with good glycemic control, the correlation became insignificant in DM patients with poor glycemic control [30]. Zhang et al. showed the non-linear “L-shaped” association between HbA1c and pulmonary function in type 2 DM [30]. Similar to a previous study, we also showed that the increased baseline and longitudinal changes of HbA1c were significantly correlated with reduced baseline and longitudinal changes of FVC in the multivariable models adjusted by age, body mass index, and laboratory variables. Furthermore, 1 L decline in FVC over 3.9 years could increase the risk of newly diagnosed type 2 DM in the present cohort study by 37.5%.

The deterioration of lung function, including FEV1 and FVC, was found in patients with type 2 DM [31]. High HbA1C and uncontrolled glycemic state were associated with the decline of FEV1 and FVC in diabetic patients [31]. In the non-diabetic participants in Korea, Oh et al. showed that increased HbA1c was significantly correlated with lung impairment, including FEV1, FVC, and restrictive pulmonary pattern [32,33]. Smoking status could influence HbA1C, FEV1 and restrictive lung pattern, as shown in previous studies [33,34]. Differently from Oh’s study (smoking 36%), our present study showed a non-significant relationship with smoking, HbA1c, and FEV1. Our study result could be affected by participants with a relatively low percentage of smoking (25.6%).

The mechanism of pulmonary dysfunction in hyperglycemic status and overt DM was multifactorial [35], including insulin resistance, oxidative stress, elastic fibers remodeling, excessive pulmonary collagens expression, and smooth muscle dysfunction [36,37,38,39]. Singh et al. showed hyperinsulinemia-induced activation of β-catenin mediated pulmonary fibrosis and epithelial-mesenchymal transition in mice and human airway smooth muscle cells [36]. Cazzola et al. demonstrated hyperglycemia-induced hyper-responsiveness of isolated human bronchi and increased intra-cellular calcium release via a Rho/ROCK signal transduction pathway in human airway smooth muscle cells [39]. Clemmer et al. showed hyperglycemia was associated with increased pulmonary vascular permeability via the superoxide pathway in obese rats [37]. Südy et al. also found that increased pulmonary tissue viscoelasticity was associated with the alveolar collapse in diabetic rats [38]. The hyperglycemic effects on the airway and alveolar regions could result in an increase in pulmonary elastic fiber remodeling, and therefore, reduce pulmonary volume [36,37,38,39,40].

### Study Limitations

There were several limitations in our longitudinal cohort study. First, air pollution and exercise habits were associated with pulmonary function [41,42]. There was no such data in the Taiwan Biobank. Second, compared to COPD patients with non-hyperlipidemia, COPD patients with hyperlipidemia were correlated with lower pulmonary volumes and higher FEV1 [43]. Although we adjusted the lipid profiles in multivariable analyses, we did not know the diary habit and lipid-lowering agents of our participants. Third, owing to the statistics of the Taiwan Biobank, the proportion of study participants coming back to follow up was only about 50%, which may result in sample bias. Thus, unequally followed participants could affect the interpretation of our results in the non-randomized study. Finally, inflammatory markers and insulin-resistant profiles were correlated with pulmonary function [44,45]. Such information was also lacking in this study.

## 5. Conclusions

During a mean 3.9-year follow-up, our present study, including participants without type 2 DM, hypertension, and COPD, demonstrated that the baseline and longitudinal changes of FVC were negatively and respectively correlated with the baseline and longitudinal change of HbA1c. Furthermore, compared to those without new-onset DM, participants with new-onset DM had a more pronounced decline of FVC over time.

## Figures and Tables

**Table 1 jpm-11-00994-t001:** Clinical, laboratory, and pulmonary function data in all 7055 study participants.

Parameters	Baseline	Follow-Up	*p*-Value	Longitudinal Change
Age (year)	49.5 ± 10.1	53.4 ± 10.0	<0.001	3.9 ± 1.3
Systolic blood pressure (mmHg)	114 ± 16	121 ± 18	<0.001	8 ± 14
Body mass index (kg/m^2^)	23.7 ± 3.4	24.0 ± 3.5	<0.001	0.3 ± 1.3
Fasting blood glucose (g/dL)	91.8 ± 7.8	93.1 ± 11.5	<0.001	1.2 ± 10.3
HbA1c (%)	5.56 ± 0.34	5.70 ± 0.44	<0.001	0.14 ± 0.36
Creatinine (mg/dL)	0.72 ± 0.28	0.71 ± 0.31	0.262	0.00 ± 0.13
GPT (u/L)	22.8 ± 18.0	22.9 ± 19.3	0.608	0.1 ± 20.9
Total cholesterol (mg/dL)	195.6 ± 34.5	198.1 ± 35.2	<0.001	2.5 ± 29.1
Triglyceride (mg/dL)	109.8 ± 81.3	114.9 ± 82.2	<0.001	5.1 ± 70.9
Pulmonary function test				
FVC (L)	2.89 ± 0.78	2.68 ± 0.76	<0.001	−0.21 ± 0.33
FVC-predicted (%)	108.1 ± 20.0	114.6 ± 22.8	<0.001	6.5 ± 17.3
FEV1 (L)	2.42 ± 0.67	2.33 ± 0.68	<0.001	−0.09 ± 0.37
FEV1-predicted (%)	111.8 ± 21.6	112.1 ± 29.7	<0.001	0.3 ± 27.8
FEV1/FVC (%)	83.8 ± 6.2	87.4 ± 9.9	<0.001	3.6 ± 10.9

FVC, forced vital capacity; FEV1, forced expiratory volume in 1 s; GPT, glutamic-pyruvic transaminase; HbA1c, glycated hemoglobin.

**Table 2 jpm-11-00994-t002:** Univariable and multivariable linear regression analyses for baseline HbA1c.

	Baseline HbA1c			
	Univariable Analysis		Multivariable Analysis	
Baseline Parameters	Unstandardized Coefficient β (95% CI)	*p*-Value	Unstandardized Coefficient β (95% CI)	*p*-Value
Age (year)	0.010 (0.009, 0.011)	<0.001	0.005 (0.005, 0.006)	<0.001
Male (%)	0.036 (0.019, 0.052)	<0.001	-	
Smoking (%)	0.016 (−0.002, 0.034)	0.088		
Systolic blood pressure (mmHg)	0.004 (0.003, 0.004)	<0.001	-	
Body mass index (kg/m^2^)	0.019 (0.017, 0.022)	<0.001	0.011 (0.009, 0.013)	<0.001
Fasting blood glucose (g/dL)	0.017 (0.016, 0.018)	<0.001	0.014 (0.013, 0.015)	<0.001
Creatinine (mg/dL)	0.023 (−0.004, 0.051)	0.100		
GPT (u/L)	0.002 (0.002, 0.003)	<0.001	0.001 (<0.001, 0.001)	<0.001
Total cholesterol (mg/dL)	0.002 (0.002, 0.003)	<0.001	0.001 (<0.001, 0.001)	<0.001
Triglyceride (mg/dL)	0.001 (<0.001, <0.001)	<0.001	<0.001 (<0.001, <0.001)	0.008
Pulmonary function test				
FVC (L)	−0.046 (−0.056, −0.036)	<0.001	−0.033 (−0.043, −0.024)	<0.001
FEV1 (L)	−0.053 (−0.065, −0.042)	<0.001	-	
FEV1/FVC (%)	−0.001 (−0.002, 0.001)	0.334	-	

CI: confidence interval; other abbreviations as in Table 1.

**Table 3 jpm-11-00994-t003:** Univariable and multivariable linear regression analyses for longitudinal changes of HbA1c.

	Longitudinal Change of HbA1C			
	Univariable Analysis		Multivariable Analysis	
Longitudinal Changes of Parameters	Unstandardized Coefficient β (95% CI)	*p*-Value	Unstandardized Coefficient β (95% CI)	*p*-Value
Age (year)	0.011 (0.004, 0.017)	0.001	-	
Systolic blood pressure (mmHg)	<0.001 (−0.001, 0.001)	0.972		
Body mass index (kg/m^2^)	0.037 (0.030, 0.043)	<0.001	0.027 (0.022, 0.033)	<0.001
Fasting blood glucose (g/dL)	0.017 (0.016, 0.017)	<0.001	0.016 (0.016, 0.017)	<0.001
Creatinine (mg/dL)	−0.142 (−0.207, −0.076)	<0.001	−0.173 (−0.231, −0.116)	<0.001
GPT (u/L)	0.001 (<0.001, 0.001)	<0.001	-	
Total cholesterol (mg/dL)	0.001 (<0.001, 0.001)	<0.001	<0.001 (<0.001, 0.001)	0.023
Triglyceride (mg/dL)	<0.001 (<0.001, <0.001)	<0.001	<0.001 (<0.001, 0.001)	0.042
Pulmonary function test				
FVC (L)	−0.032 (−0.058, −0.007)	0.013	−0.025 (−0.048, −0.003)	0.026
FEV1 (L)	−0.031 (−0.053, −0.008)	0.008	-	
FEV1/FVC (%)	<0.001 (−0.001, 0.001)	0.764	-	

CI: confidence interval; other abbreviations as in Table 1.

**Table 4 jpm-11-00994-t004:** Univariable and multivariable linear regression analyses for longitudinal changes of FVC.

	Longitudinal Change of FVC			
	Univariable Analysis		Multivariable Analysis	
Longitudinal Changes of Parameters	Unstandardized Coefficient β (95% CI)	*p*-Value	Unstandardized Coefficient β (95% CI)	*p*-Value
Age (year)	−0.017 (−0.023, −0.011)	<0.001	−0.017 (−0.023, −0.011)	<0.001
Smoking (baseline)	−0.059 (−0.077, −0.042)	<0.001	−0.059 (−0.077, −0.042)	<0.001
Systolic blood pressure (mmHg)	−0.001 (−0.001, <0.001)	0.228		
Fasting blood glucose (g/dL)	0.001 (−0.001, 0.001)	0.637		
HbA1C (%)	−0.027 (−0.049, −0.006)	0.013	−0.022 (−0.044, −0.001)	0.041
Creatinine (mg/dL)	0.006 (−0.054, 0.060)	0.842		
GPT (u/L)	−0.001 (−0.001, <0.000)	0.048	−0.001 (−0.001, <0.000)	0.035
Total cholesterol (mg/dL)	0.001 (<−0.001, 0.001)	0.950		
Triglyceride (mg/dL)	−0.001 (<−0.001, 0.001)	0.051		

CI: confidence interval; other abbreviations as in Table 1.

**Table 5 jpm-11-00994-t005:** Comparison of longitudinal changes of clinical, laboratory, and pulmonary function data between participants with and without newly diagnosed type 2 diabetes mellitus.

Longitudinal Changes of Parameters	With Newly Diagnosed Type 2 Diabetes Mellitus n = 271 (3.8%)	Without Newly Diagnosed Type 2 Diabetes Mellitus N = 6784 (96.2%)	*p*-Value
Age (year)	4.1 ± 1.3	3.9 ± 1.3	0.086
Systolic blood pressure (mmHg)	9 ± 15	8 ± 14	0.193
Body mass index (kg/m^2^)	0.4 ± 1.3	0.3 ± 1.3	0.159
Fasting blood glucose (g/dL)	16.6 ± 34.8	0.6 ± 7.1	<0.001
HbA1c (%)	0.92 ± 0.95	0.11 ± 0.27	<0.001
Creatinine (mg/dL)	0.0 ± 0.1	0.0 ± 0.1	0.167
GPT (u/L)	1.8 ± 21.4	0.0 ± 20.1	0.187
Total cholesterol (mg/dL)	−5.7 ± 40.1	2.8 ± 28.6	<0.001
Triglyceride (mg/dL)	−0.2 ± 120.7	5.3 ± 68.1	0.211
Pulmonary function test			
FVC (L)	−0.25 ± 0.29	−0.21 ± 0.33	0.028
FVC-predicted (%)	8.3 ± 14.8	6.4 ± 17.4	0.071
FEV1 (L)	−0.12 ± 0.31	−0.09 ± 0.37	0.136
FEV1-predicted (%)	0.0 ± 17.8	0.3 ± 28.1	0.949
FEV1/FVC (%)	3.9 ± 9.0	3.6 ± 10.9	0.613

Abbreviations as in Table 1.

**Table 6 jpm-11-00994-t006:** Odds ratio of longitudinal changes of parameters in association with newly diagnosed type 2 DM in univariable and multivariable logistic regression analyses.

	Newly Diagnosed Type 2 Diabetes Mellitus			
	Univariable Analysis		Multivariable Analysis	
Longitudinal Changes of Parameters	Odds Ratio (95% CI)	*p*-Value	Odds Ratio (95% CI)	*p*-Value
Age (per 1 year)	1.084 (0.989, 1.118)	0.086		
Systolic blood pressure (per 1 mmHg)	1.006 (0.997, 1.015)	0.193		
Body mass index (per 1 kg/m^2^)	1.068 (0.976, 1.169)	0.155		
Fasting blood glucose (per 1 g/dL)	1.102 (1.089, 1.115)	<0.001	1.103 (1.090, 1.117)	<0.001
Creatinine (per 1 mg/dL)	0.379 (0.114, 1.261)	0.114		
GPT (per 1 u/L)	1.003 (0.999, 1.007)	0.180		
Total cholesterol (per 1 mg/dL)	0.991 (0.987, 0.994)	<0.001	0.989 (0.985, 0.993)	<0.001
Triglyceride (per 1 mg/dL)	0.999 (0.997, 1.001)	0.204		
Pulmonary function test				
FVC (per 1 L)	0.669 (0.467, 0.957)	0.028	0.625 (0.424, 0.922)	0.018
FEV1 (L) (per 1 L)	0.791 (0.581, 1.076)	0.136		
FEV1/FVC (per 1%)	1.003 (0.992, 1.014)	0.612		

CI: confidence interval; other abbreviations as in Table 1.

## Data Availability

Data may be available upon request to interested researchers. Please send data requests to: Ho-Ming Su, MD. Division of Cardiology, Department of Internal Medicine, Kaohsiung Medical University Hospital, Kaohsiung Medical University.

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
