# Peer review of "Association of Pulmonary Function Decline over Time with Longitudinal Change of Glycated Hemoglobin in Participants without Diabetes Mellitus"

_jpm, 2021, doi:10.3390/jpm11100994_

Round 1

Reviewer 1 Report

GENERAL COMMENTS

I read with interest the article by Lee and colleagues entitled “Association of pulmonary function decline over time with the longitudinal change of glycated hemoglobin in participants without diabetes mellitus” submitted for publication in the Journal of Personalized Medicine. 

In this interesting longitudinal study, the authors describe for the first time that baseline and longitudinal change of forced vital capacity (FVC) were negatively and respectively correlated with baseline and longitudinal change of HbA1c in subjects without type 2 diabetes mellitus (DM), hypertension, and COPD. However, there are several methodological problems, that should be solved and included in the manuscript before considering their publication.

The first is that not enough data are provided in the Methods section or in the cited references (13,14) to indicate that it is a population-based study. In principle, it seems that the population is selected from the Taiwan Biobank Database in adult patients without cancer and if there is no random selection of the subjects in said study, it cannot be said of as a population-based study, since the sample, although large, it would not necessarily have to be representative of the population of Taiwan. The authors should clarify this aspect.

Second, the authors should provide more data from the longitudinal study. For example, when we're spirometries performed during follow-up? How many patients with two spirometries (at baseline and end of follow-up) had longitudinal tests of HbA1c?

Finally, as smoking is one of the factors that most influence the loss of lung function (and 25.6% of the subjects included were smokers at baseline), it is not understood why this variable is not included in the statistical analysis related to the longitudinal change of HbA1c (Tables 3 and 4). Although there is no significant association between baseline smoking and baseline HbA1, without this information (baseline smoking and better also longitudinal smoking changes), it cannot be ruled out that the decrease in FVC is actually related to smoking and not to changes in HbA1c.

On the other hand, most of the previous studies in patients with type2-DM found a relationship between HBA1c and FEV1, which did not appear in the present study. The authors should explain these discrepancies in the discussion section.

SPECIFIC COMMENTS

Page 1, line 2 (Abstract). There is a typo: diabetes instead of diabetes.

Page 2, line 11 of subheading “Data Source”. There is a typo: Tests instead of testes

Page 3, line 2. There is a space left after the reference [16]

Author Response

Dear Editor:

Thank you for the thorough review on our manuscript (reference number: 1361139), entitled " Association of pulmonary function decline over time with longitudinal change of glycated hemoglobin in participants without diabetes mellitus". All the comments from the editor and reviewers are carefully considered, and the manuscript is revised according to the comments. We appreciate the reviewers’ kind instructions, suggestions, and corrections. For contrast, the corrections or additions are highlighted in red words in Microsoft Word.

Sincerely yours,

Ho-Ming Su, MD, E-mail: cobeshm@seed.net.tw

Cardiology/Internal Medicine, Kaohsiung Medical university, 100 Shih-Chuan 1st Road, Kaohsiung 807, Taiwan

Fax: (886) (7) 323-4845

The comments of the reviewer 1 are as follows:

Review 1 comment

GENERAL COMMENTS

I read with interest the article by Lee and colleagues entitled “Association of pulmonary function decline over time with the longitudinal change of glycated hemoglobin in participants without diabetes mellitus” submitted for publication in the Journal of Personalized Medicine. 

In this interesting longitudinal study, the authors describe for the first time that baseline and longitudinal change of forced vital capacity (FVC) were negatively and respectively correlated with baseline and longitudinal change of HbA1c in subjects without type 2 diabetes mellitus (DM), hypertension, and COPD. However, there are several methodological problems, that should be solved and included in the manuscript before considering their publication.

  1. The first is that not enough data are provided in the Methods section or in the cited references (13,14) to indicate that it is a population-based study. In principle, it seems that the population is selected from the Taiwan Biobank Database in adult patients without cancer and if there is no random selection of the subjects in said study, it cannot be said of as a population-based study, since the sample, although large, it would not necessarily have to be representative of the population of Taiwan. The authors should clarify this aspect.
  • Thanks for your great comment. Indeed, the study design of Taiwan Biobank was not a population-based study and could be selection and sampling bias. We revised the study design in the Methods section and Study limitation. (Revised manuscript, Page 2, Subheading “Data Source”: Line 1)
  1. Second, the authors should provide more data from the longitudinal study. For example, when we're spirometries performed during follow-up? How many patients with two spirometries (at baseline and end of follow-up) had longitudinal tests of HbA1c?
  • Thanks for your great comment. Participants follow up at every 2 to 4 years after fist enrolled in the Taiwan Biobank study. In our present study, 7055 participants had two spirometry examination and serum laboratory test. However, owing to the statistics of Taiwan Biobank, the proportion of study participants coming back to follow up was only about 50%, which may be result in sample bias. Thus unequally followed participants could be affect the interpretation of our results in the non-randomized study. We add the statement in the Limitation section. (Revised manuscript, Page 8, Subheading Study Limitations, Line 6-9)
  1. Finally, as smoking is one of the factors that most influence the loss of lung function (and 25.6% of the subjects included were smokers at baseline), it is not understood why this variable is not included in the statistical analysis related to the longitudinal change of HbA1c (Tables 3 and 4). Although there is no significant association between baseline smoking and baseline HbA1, without this information (baseline smoking and better also longitudinal smoking changes), it cannot be ruled out that the decrease in FVC is actually related to smoking and not to changes in HbA1c.
  • Thanks for your very great comment. Smoking is an important factor in the pulmonary function decline. Longitudinal change of FVC was statistical analysis by baseline smoking and clinical parameters in Table 4. Univariable and multivariable linear regression analysis of longitudinal change of FVC are showed in Table 4. In univariable and multivariable analysis, longitudinal change of FVC was negatively correlated with smoking, and longitudinal changes of age, GPT, and HbA1c (in multivariable analysis, unstandardized coefficient β = -0.022, 95% CI, -0.044 to -0.001, P = 0.041). (Revised manuscript, Page 5, Table 4, Line 1-5)
  1. On the other hand, most of the previous studies in patients with type2-DM found a relationship between HBA1c and FEV1, which did not appear in the present study. The authors should explain these discrepancies in the discussion section.
  • Thanks for your great comment. Deterioration of lung function included FEV1 and FVC were found in patients with type 2 DM. High HbA1C and uncontrolled glycemic state were associated with decline of FEV1 and FVC in diabetic patients. In the non-diabetic participants in Korea, Oh et al. showed that increased HbA1c was significantly correlated with lung impairment included FEV1, FVC, and restrictive pulmonary pattern. Smoking status could influence HbA1C, FEV1 and restrictive lung pattern in previous studies. Unlike Oh’s study (smoking 36%), our present study showed the non-significantly relationship within smoking, HbA1c, and FEV1. Our study result could be affected by participants with relatively low percentage of smoking (25.6%). We mentioned that in the Discussion section. (Revised manuscript, Discussion section, Page 7, Line 50-51; Page 8, Line 1-5)

SPECIFIC COMMENTS

  1. Page 1, line 2 (Abstract). There is a typo: diabetes instead of diabetes.
  • Thanks for your great comment. We corrected the type error. (Revised manuscript, Page 1, Line 2)
  1. Page 2, line 11 of subheading “Data Source”. There is a typo: Tests instead of testes
  • Thanks for your great comment. We corrected the type error. (Revised manuscript, Page 2, Line 11)
  1. Page 3, line 2. There is a space left after the reference [16]
  • Thanks for your great comment. We corrected the type error. (Revised manuscript, Page 3, Line 2)

Reviewer 2 Report

The work of Lee at al. is based on a large database, and the study presents a new aspect of the relationship between decline in lung function and HbA1c. I appreciate the work done and recommend the publication of this work taking into account the following two points.

First, it is not entirely clear from the results what the relationship is between worsening lung function and other parameters, particularly fasting blood glucose and triglycerides. The authors present the correlation between HbA1c and other parameters, but it is not clear to what extent the decline in lung function correlates directly with these other parameters.

Secondly, the authors write in the "study limitations" that they have no data on lipid-lowering agents in the study participants. It should be noted that this is an important point. Therefore, I would recommend adding at least some estimates of the percentage of people of this age taking lipid and glucose lowering agents.

Author Response

Dear Editor:

Thank you for the thorough review on our manuscript (reference number: 1361139), entitled " Association of pulmonary function decline over time with longitudinal change of glycated hemoglobin in participants without diabetes mellitus". All the comments from the editor and reviewers are carefully considered, and the manuscript is revised according to the comments. We appreciate the reviewers’ kind instructions, suggestions, and corrections. For contrast, the corrections or additions are highlighted in red words in Microsoft Word.

Sincerely yours,

Ho-Ming Su, MD, E-mail: cobeshm@seed.net.tw

Cardiology/Internal Medicine, Kaohsiung Medical university, 100 Shih-Chuan 1st Road, Kaohsiung 807, Taiwan

Fax: (886) (7) 323-4845

The comments of the reviewer 2 are as follows:

The work of Lee at al. is based on a large database, and the study presents a new aspect of the relationship between decline in lung function and HbA1c. I appreciate the work done and recommend the publication of this work taking into account the following two points.

  1. First, it is not entirely clear from the results what the relationship is between worsening lung function and other parameters, particularly fasting blood glucose and triglycerides. The authors present the correlation between HbA1c and other parameters, but it is not clear to what extent the decline in lung function correlates directly with these other parameters.
  • Thanks for your great comment. We studied parameters on decline of FVC and consider baseline smoking status. Longitudinal change of FVC was statistical analysis by baseline smoking and clinical parameters in Table 4. Univariable and multivariable linear regression analysis of longitudinal change of FVC are showed in Table 4. In univariable and multivariable analysis, longitudinal change of FVC was negatively correlated with smoking, and longitudinal changes of age, GPT, and HbA1c (in multivariable analysis, unstandardized coefficient β = -0.022, 95% CI, -0.044 to -0.001, P = 0.041). (Revised manuscript, Page 5, Table 4, Line 1-5)
  1. Secondly, the authors write in the "study limitations" that they have no data on lipid-lowering agents in the study participants. It should be noted that this is an important point. Therefore, I would recommend adding at least some estimates of the percentage of people of this age taking lipid and glucose lowering agents.
  • Thanks for your great comment. Hyperlipidemia and its related lipid lowing agents were associated with pulmonary function. In Taiwan Biobank, because of participants’ medical characteristics obtained from face-to-face interview and questionnaire, how many and what kinds of participants’ medication were lack from the Databank. The percentage of participants took lipid-lowing medication was really lack in Taiwan Biobank. Therefore, initially we enrolled participants without diabetes, hypertension and pulmonary diseases. The influences of lipid profiles were also considered in the statistical analysis. Owing to impact of lipid lowing agents on pulmonary function, we wrote it in the study limitation.
